# A Full Set of In Vitro Assays in Chitosan/Tween 80 Microspheres Loaded with Magnetite Nanoparticles

**DOI:** 10.3390/polym13030400

**Published:** 2021-01-27

**Authors:** Jorge A Roacho-Pérez, Kassandra O Rodríguez-Aguillón, Hugo L Gallardo-Blanco, María R Velazco-Campos, Karla V Sosa-Cruz, Perla E García-Casillas, Luz Rojas-Patlán, Margarita Sánchez-Domínguez, Ana M Rivas-Estilla, Víctor Gómez-Flores, Christian Chapa-Gonzalez, Celia N Sánchez-Domínguez

**Affiliations:** 1Departamento de Bioquímica y Medicina Molecular, Facultad de Medicina, Universidad Autónoma de Nuevo León, Monterrey 64460, Mexico; jorge.roacho@uacj.mx (J.A.R.-P.); kora_g416@hotmail.com (K.O.R.-A.); ana.rivasst@uanl.edu.mx (A.M.R.-E.); 2Departamento de Genética, Facultad de Medicina, Universidad Autónoma de Nuevo León, Monterrey 64460, Mexico; hugoleonid2011@icloud.com (H.L.G.-B.); roble.velazco@gmail.com (M.R.V.-C.); qcb.luz.rojas@gmail.com (L.R.-P.); 3Instituto de Ingeniería y Tecnología, Universidad Autónoma de Ciudad Juárez, Ciudad Juárez 32310, Mexico; ksosacruz@yahoo.com (K.V.S.-C.); pegarcia@uacj.mx (P.E.G.-C.); victor.gomez@uacj.mx (V.G.-F.); 4Centro de Investigación en Materiales Avanzados, S.C. (CIMAV, S.C.), Unidad Monterrey, Apodaca 66628, Mexico; margarita.sanchez@cimav.edu.mx

**Keywords:** polymers, magnetite nanoparticles, chitosan, Tween 80, synthesis, nanotoxicology, genotoxicity, hemotoxicity

## Abstract

Microspheres have been proposed for different medical applications, such as the delivery of therapeutic proteins. The first step, before evaluating the functionality of a protein delivery system, is to evaluate their biological safety. In this work, we developed chitosan/Tween 80 microspheres loaded with magnetite nanoparticles and evaluated cell damage. The formation and physical–chemical properties of the microspheres were determined by FT-IR, Raman, thermogravimetric analysis (TGA), energy-dispersive X-ray spectroscopy (EDS), dynamic light scattering (DLS), and SEM. Cell damage was evaluated by a full set of in vitro assays using a non-cancerous cell line, human erythrocytes, and human lymphocytes. At the same time, to know if these microspheres can load proteins over their surface, bovine serum albumin (BSA) immobilization was measured. Results showed 7 nm magnetite nanoparticles loaded into chitosan/Tween 80 microspheres with average sizes of 1.431 µm. At concentrations from 1 to 100 µg/mL, there was no evidence of changes in mitochondrial metabolism, cell morphology, membrane rupture, cell cycle, nor sister chromatid exchange formation. For each microgram of microspheres 1.8 µg of BSA was immobilized. The result provides the fundamental understanding of the in vitro biological behavior, and safety, of developed microspheres. Additionally, this set of assays can be helpful for researchers to evaluate different nano and microparticles.

## 1. Introduction

Conventional methods for drug administration involve the use of tablets, capsules, and injections [1]. Although these methods are functional for the treatment of several health conditions, the complexity of human diseases has led scientists to develop new strategies for drug administration. One of the tissues with difficult access is the brain. Some disorders of the central nervous system such as stroke, brain tumors, and neurodegenerative diseases can be treated with therapeutic proteins. The delivery of these proteins is complicated because blood–brain barriers protect the brain from the entrance of outside substances [2].

Microspheres have been studied as a vehicle for the delivery of therapeutic agents [3]. Polymeric microspheres increase the therapeutic agents’ useful lifetime because they protect the therapeutic agent from enzymatic degradation [4]. Chitosan microspheres have been studied and developed by several research groups as a hemostatic [5], antibacterial [6], and anti-inflammatory agent [7], and a vehicle for drug [8] and vaccine delivery [9]. Chitosan (CS), obtained after alkaline deacetylation of chitin, is a pH-sensitive natural polycationic polysaccharide. Some attractive properties of chitosan are its low cost, cationic charge, biocompatibility, and chemical stability [10]. Chitosan combined with magnetite nanoparticles have a broad field of application, but mainly they have been the subject of research in the supply of drugs, proteins, peptides, genes, DNA, and others [11,12,13]. Magnetic nanoparticles can target the microsphere to a specific tissue, using an external magnetic field [14]. Some magnetic nanomaterials have reached clinical trials, and others are available in the market for their use in humans [15,16,17,18,19]. The most used magnetic nanoparticles are iron oxides, such as magnetite (Fe_3_O_4_), because of its high degree of biocompatibility and unique magnetic properties that can be controlled by the synthesis methods [20]. Another material that can be used for the formation of the microspheres is the non-ionic surfactant polysorbate 80 (Tween^®^ 80). The novelty of Tween 80 is attributed to its ability to be targeted into the brain after intravenous injection. Tween 80 binds to the plasma lipoprotein Apo-E, which attaches to LDL receptors on brain microvascular endothelial cells. Therefore, Tween 80 can deliver pharmaceutical substances into the brain [21,22].

Before testing the functionality of microspheres, it is essential to develop and apply in vitro tests that can predict the potential toxicity of particles. It is fundamental to evaluate the biological safety of particles to ensure their biocompatibility, especially because different nanoparticle synthesis methodologies generate different physical properties that lead to distinct biological behaviors. The toxicity of nanoparticles occurs at molecular and cellular levels. When particles move through the body, they interact with different biological environments such as blood, extracellular matrix, cytoplasm, cell organelles, and nucleus. Consequently, this interaction can affect some biomolecules and cellular components [23]. The exposed biomolecules can suffer structural and functional alterations [24,25]. Nanoparticles can also cause cytotoxicity by breaking several membranes outside and inside the cell. When the membrane integrity is compromised, the content of the membrane compartments can leak. This leak can cause cell stress and interfere with the normal function of the cell and organelles [26,27].

Although several papers evaluate the cytotoxicity of chitosan microspheres, different synthesis methodologies generate different physical properties that interfere with the biological behavior of microspheres. The main objective of this work was to determine if chitosan/Tween 80 microspheres loaded with magnetite nanoparticles can cause cell damage. We evaluated cell metabolic activity (MTT assay), morphological cell changes (H&E staining), cell lysis of membrane (hemolysis test), chromosome changes (SCE test), and evaluation of the cell cycle (lymphocyte culture).

## 2. Materials and Methods

The Ethics in the Research Committee of the School of Medicine and Dr. José Eleuterio Gonzalez University Hospital of the Universidad Autónoma de Nuevo León reviewed and approved this methodology in September 2017, with the project identification code BI17-00001. All human donors were treated according to ethical standards.

### 2.1. Chitosan/Tween 80 Microspheres Loaded with Magnetite Nanoparticles Preparation

Magnetite nanoparticles were synthetized by chemical coprecipitation based on the methodology used in this research previously reported [28]. The preparation of the microspheres is shown in Figure 1. A sample of 90 mg of magnetite nanoparticles was dispersed in 45 mL of a 7% Tween 80 solution (Sigma-Aldrich, St. Louis, MO, USA) using an ultrasonic processor (Fisher Scientific, Pittsburgh, PA, USA) amplitude 80%. A total of 90 mg of medium molecular weight chitosan (Sigma-Aldrich, St. Louis, MO, USA) was dissolved until homogenization in 45 mL of a 2% glacial acetic acid solution (Thermo Fisher Scientific, Waltham, MA, USA). Chitosan solution was added drop by drop into the magnetite nanoparticles solution and mixed with an ultrasonic processor. In order to induce the protonation of the chitosan, the pH of the mix was adjusted to 5.5 with ammonium hydroxide (NH_4_OH, CTR Scientific, Monterrey, Mexico). Magnetite-CS/Tween 80 microspheres were isolated by centrifugation at 14,000 rpm for 10 min and washed with distilled water repeatedly until magnetite-CS/Tween 80 microspheres reached pH 7. Magnetite-CS/Tween 80 microspheres were dried at 50 °C for 24 h and pulverized in an agate mortar and stored.

### 2.2. Nanoparticle Characterization

Functional group analysis was evaluated by Fourier transform infrared spectroscopy (FT-IR NICOLET 6700, Thermo Fisher Scientific, Waltham, MA, USA), and Raman spectra were obtained in an instrument equipped with a 785 nm laser source (Alpha 300 RA, WITec, Ulm, Germany). While, thermogravimetric analysis from room temperature to 700 °C was accomplished on a SDT 2960 Simultaneous DSC-TGA (TA Instrument, New Castle, DE, USA). The samples’ morphology was examined with scanning electron microscopy (SEM JSM-7000F, JEOL, Akrishima, Japan). For SEM analysis, the different samples were dispersed in distilled water and then spread on a carbon tape slide. Once the sample had dried, a secondary electron image (SEI) and energy-dispersive X-ray spectroscopy (EDS) study were performed. The samples’ distribution size was measured with dynamic light scattering (DLS) equipment (Nanotrack Wave II, Microtrac, Haan, Germany). In order to avoid nanoparticle agglomeration, nanoparticle samples were prepared immediately before their analysis [29]. For DLS analysis, nanoparticles and microspheres were dispersed (1 mg/mL) into previously filtered distilled water.

### 2.3. Cytotoxicity: MTT Assay and H&E Staining

For the measurement of the microspheres’ capacity to cause cell death a 3-(4,5-dimethylthiazol-2-yl)-2,5-diphenyltetrazolium bromide (MTT) assay and a hematoxylin–eosin (H&E) staining was developed using the embryo mouse fibroblast cell line 3T3L1 (ATCC, Manassas, VA, USA). A total of 10,000 cells were seeded per well, in a 96-well microplate, and cultured with supplemented DMEM. The supplemented DMEM medium contained Dulbecco’s modified Eagle’s medium (Thermo Fisher Scientific, Waltham, MA, USA), 10% of fetal bovine serum (Thermo Fisher Scientific, Waltham, MA, USA), and 1% of penicillin-streptomycin (Thermo Fisher Scientific, Waltham, MA, USA). After 24 h, cells were exposed to the microspheres previously sterilized by 15 min of UV light exposure. The microspheres were evaluated at different concentrations by triplicate from 1 to 10,000 µg/mL. Cells without exposition to any particle were taken as a negative control. Cell death was measured after 24, 48, and 72 h of exposure.

For the MTT assay, cells were processed with the Cell Proliferation Kit I (Roche, Basilea, Switzerland) following the instructions. Absorbance was read by UV–vis spectroscopy (Nanodrop, Thermo Fisher Scientific, Waltham, MA, USA) at 570 nm. For the results analysis, the negative control absorbance was adjusted as 100% of cell viability. The calculation of cell viability of the treated cells was calculated according to the negative control. Thereafter H&E staining was performed. The medium was removed, and cells were washed with PBS (Thermo Fisher Scientific, Waltham, MA, USA) three times. For the fixation of the cells, each well was incubated 10 min at −20 °C with 50 µL of cold methanol. Cells were rewashed with PBS three times. For the stain, cells were incubated 5 min in hematoxylin, tap water wash, HCl (diluted at 0.5% in ethanol) wash, tap water wash, distilled water wash, 5 min in eosin, and a tap water wash. Morphology of the cells was analyzed in an optical microscope (Inverted microscopeCKX41, Olympus, Shinjuku, Japan).

### 2.4. Hemolysis Test

A hemolysis test was developed to determine the effect of microspheres on erythrocyte lysis. A heparinized tube (Becton Dickinson, Franklin Lakes, NJ, USA) with blood from a healthy donor was used. Erythrocytes were isolated by centrifugation (3000 rpm for 4 min) and washed three times with Alsever’s solution (dextrose 0.116 M, sodium chloride 0.071 M, sodium citrate 0.027 M, and citric acid 0.002 M, pH 6.4). Microspheres were dispersed in Alsever’s solution at different concentrations (1–10,000 µg/m). Quintupled samples of microspheres were incubated with the erythrocytes in a relation of 1:99 erythrocytes: microspheres v/v. The incubation conditions were 37 °C in agitation at 400 rpm for 30 min. The positive control was a suspension of erythrocytes in distilled water. Negative control was a suspension of erythrocytes in Alsever’s solution. After incubation, samples were centrifuged (3000 rpm for 4 min), and the hemoglobin released in the supernatant was measured by UV–Vis spectroscopy at 415 nm. The absorbance of the positive control was adjusted as 100% of hemolysis, and each of the samples were calculated according to the positive control.

### 2.5. Sister Chromatid Exchange Assay

For this analysis, a primary culture of human lymphocytes was used. Blood from a healthy donor was collected in a heparinized tube. An aliquot of 500 µL of blood was cultured in 5 mL of RPMI-1640 culture medium (Thermo Fisher Scientific, Waltham, MA, USA), 100 µL of phytohemagglutinin for mitosis induction (Thermo Fisher Scientific, Waltham, MA, USA), and microspheres at different concentrations (from 1 to 100 µg/mL). For the positive control, mitomycin C 40 ng/mL (Sigma-Aldrich, St. Louis, MO, USA) was added. After 24 h of culture, bromodeoxyuridine, a synthetic thymidine analog nucleoside, (BrdU, Sigma-Aldrich, St. Louis, MO, USA) was added to the medium at a final concentration of 23.1 µM. In order to stop mitosis in metaphase, after 48 h of incubation with BrdU, KaryoMax Colcedim were added to the medium at a final concentration of 10 µg/mL (Thermo Fisher Scientific, Waltham, MA, USA). After 40 min of incubation with colcedim, cells were centrifugated for 10 min at 1200 rpm. Supernatant was eliminated and cells were incubated with a hypotonic solution of KCl 0.075 M for 20 min, and were washed with a solution of acetic acid and methanol (1:3, v/v) until the precipitate turned white. The cell suspension was dropped into a clean microscope slide for fixation. Cells over the slides were incubated for 40 min with bisbenzimide (Sigma-Aldrich, St. Louis, MO, USA) to stain A-T rich regions of the chromosome. After this incubation, samples were exposed to UV light for 105 min and covered with a saline-sodium citrate buffer 0.5× for 105 min. Finally, samples were covered with Giemsa stain at 2% in PBS, pH 6.8. Samples were dried and analyzed in an optical microscope. Segments of genetic material exchanged from one chromatid to another were counted in 50 metaphases per sample.

### 2.6. Surface Protein Load

Bovine serum albumin (BSA) was used as a model molecule for measuring protein load over the microspheres surface. With this experiment it can be demonstrated that the synthesized microspheres can immobilize a protein over their surface. The loaded protein can be replaced with other different protein with therapeutic properties for brain diseases. Triplicated samples of 500 µg of microspheres were incubated in agitation for 30 min in 300 mL of a PBS solution (pH 7.4, room temperature) with BSA (10 different amounts from 879 to 8799 µg). After incubation, microspheres were precipitated using a magnetic plate. The non-immobilized proteins were taken from the supernatant and measured by UV–Vis spectroscopy at 280 nm.

### 2.7. Statistical Analysis

Biological assays data were studied by an analysis of variance (ANOVA) and a Tukey’s HSD (honestly significant difference) test, with a confidence interval of 95% to determine significant differences between the control group and the test samples.

## 3. Results and Discussion

### 3.1. Microsphere Characterization: FT-IR Spectroscopy

FTIR spectra of bare magnetite and chitosan/Tween 80 microspheres loaded with magnetite nanoparticles are shown in Figure 2. This characterization is based on the vibrations of the chemical bonds of the analyzed samples. It means that FTIR evidences the functional groups present in the sample. Bare magnetite nanoparticles show the main band at 547 cm^−1^, also reported by other authors, corresponding to Fe-O stretching in the octahedral site of the crystal structure of magnetite [30,31,32]. The microspheres spectrum shows more bands due to the different functional groups present in the chitosan and Tween 80. The spectrum shows bands at 3453 cm^−1^ and 1637 cm^−1^ correspond to vibrational stretching and bending modes of the hydroxyl group. The bands at 2921 cm^−1^ and 2852 cm^−1^ correspond to C-H asymmetrical and symmetrical, respectively, stretching vibrations. The band at 1267 cm^−1^ was related to the C-H bending mode. The band at 1186 cm^−1^ was related to the stretching vibration of the C-O link. The band observed at 1083 cm^−1^ corresponds to the asymmetric stretch vibration of the C-O-C bond. The band at 818 cm^−1^ corresponds to a C-H deformation. The band that corresponds only to Tween 80 is presented at 1741 cm^−1^, which indicates a C=O stretching vibration. The bands that correspond only to the chitosan are the vibratory stretching mode of C-N observed at 1468 cm^−1^ and the bending mode of N-H at 1508 cm^−1^ [33,34,35,36,37].

### 3.2. Microsphere Characterization: Raman Spectroscopy

The Raman signal obtained from the microspheres is shown in Figure 3. The strongest bands showed in the spectrum correspond to the inorganic compound. According with the Bio-Red’s HORIBA library Edition KnowItAll, the bands shown at 213.5, 277.6, 389.7, 584.4, and 1249 cm^−1^ correspond to the spectrum RMX #265 Magnetite (heated). This spectrum is the result of the changes produced in the magnetite by the heat of the power laser [38,39]. The intensity of the polymeric organic compounds is lower than the inorganic part, but some bands were identified for the bending vibrations of C-NH-C (277.6 cm^−1^), COC (469 cm^−1^), CO-NH (497 cm^−1^), C-CH_3_ (497 cm^−1^), and CH (1249 cm^−1^). Additionally, the band at 1249 cm^−1^ can be ascribed to the stretching of C-C and C-O [40,41].

### 3.3. Microsphere Characterization: TGA

Thermal analysis provides information about the polymeric composition of the sample. The results obtained from magnetite nanoparticles and microspheres of chitosan/Tween 80 loaded with magnetite nanoparticles are shown in Figure 4. Magnetite nanoparticles showed a loss of 4% of their weight, possibly because of a residual water loss. Microspheres showed a 16.7% of their weight loss at two different temperatures. Tween 80 weight loss temperature is shown at two different temperatures near 250 and 350 °C respectively [42]. The weight loss at a temperature of 300 °C is given by the degradation of the chitosan [43]. The spectra confirm the presence of both components in the sample.

### 3.4. Microsphere Characterization: SEM-EDS

To ensure that M-CS/Tween 80 microspheres have adopted the desired spherical morphology, they were observed through SEM. It is observed in Figure 5a that M-CS/Tween 80 microspheres adopt the expected spherical morphology. Images show spherical nanoparticles inside the chitosan microspheres. An EDS study was developed to ensure that the nanoparticles inside the chitosan microsphere correspond to magnetite. Figure 5b shows the distribution of Fe inside the microspheres. Fe distribution corresponds with the nanoparticles inside the microspheres, proving that magnetite nanoparticles are inside the chitosan microspheres.

### 3.5. Microspheres Characterization: DLS

The size distribution of magnetic nanoparticles (Figure 6a) and M-CS/Tween 80 microspheres (Figure 6b) was measured using the DLS technique. Both structures show sizes with a normal distribution. The average size of the magnetite nanoparticles obtained was 7 nm. This size is consistent with that reported by authors for chemical coprecipitation by the rapid injection synthesis method [28]. The average size of 98.8% of M-CS/Tween 80 microspheres was 1.431 µm, with a range size that goes from 1.431 to 1.756 µm. Only 1.2% of synthesized microspheres show average sizes of 6 µm. The size could correspond to agglomerates due to the polydispersity index (PDI) being 0.77 for magnetite nanoparticles and 0.22 for M-CS/Tween 80 microspheres.

### 3.6. Cytotoxicity of M-CS/Tween 80 Microspheres: MTT Assay

Cytotoxicity tests evaluate cell damage, growth, and specific aspects of cell metabolism. The MTT assay is based on measuring the metabolic activity of cells. MTT is a yellow and water-soluble agent; the viable cells reduce it into a blue-violet insoluble compound called formazan. The number of viable cells is correlated with the presence of formazan in the sample [44]. Evaluation of the cytotoxicity of magnetite nanoparticles and M-CS/Tween 80 microspheres was measured by exposure of 3T3L1 cells, followed by the MTT assay. The evaluation was performed after 24, 48, and 72 h of exposure. Figure 7 shows a comparison between the cytotoxic effect of magnetite nanoparticles vs. M-CS/Tween 80 microspheres. An increment in particle concentration increments cytotoxicity. An increment in time exposure also increments particle cytotoxicity. Statistical analysis (*p =* 0.05) showed that M-CS/Tween 80 microspheres are less cytotoxic in comparison with bare magnetite nanoparticles. This result demonstrates the non-toxic properties of chitosan and Tween 80. At the first 24 h of exposure, M-CS/Tween 80 microspheres showed a non-cytotoxic behavior at concentrations from 1 to 100 µg/mL. In summary, this experiment showed that the use of chitosan and Tween 80 improved the magnetite biological behavior but was only safe to use at concentrations lower than 100 µg/mL, and preferably, in short exposure times.

The results obtained by measuring the viability of 3T3L1 cells after exposure with magnetite nanoparticles and M-CS/Tween 80 microspheres at concentrations of 1–10,000 µg/mL were compared by those obtained by Lotfi et al. In their study, they used concentrations of 10, 25, 50, 75, and 100 µg/mL of magnetite nanoparticles coated with chitosan for 24 h in MCF7 cells and fibroblasts. They obtained 78% cell viability for bare magnetite nanoparticles and a cell viability of 80% for magnetite coated with chitosan in the MCF7 cell line [35]. Viabilities obtained by our research group at concentrations of 100 µg/mL at the first 24 h are 87% for magnetite nanoparticles and 93% for M-CS/Tween 80 microspheres. It is essential to evaluate the cytotoxicity of the particles. Changes in the properties of the nanoparticles are due to batch-to-batch variations, which are reflected in their interaction with the cells. Additionally, it must be considered that the results reflect the specific response of each cell line. Additionally, results obtained in this work give us a screening of particles’ interaction in 48 and 72 h. The cytotoxicity of these particles increased with an increment in time exposure. Therefore, it is recommended not to use them in applications when a long-time exposure is needed.

### 3.7. Cytotoxicity of M-CS Microspheres: H&E Staining

To observe morphological changes in the 3T3L1 cells due to their exposure with particles, H&E staining was performed. Figure 8 shows images from an optical microscope at 40× after 24 h of exposure to the particles. Cell morphology changes and a decrease in cell confluence were observed at concentrations of 1000 and 10,000 µg/mL with both types of particles. At concentrations of 1, 10, and 100 µg/mL, there were no evident changes in cytoplasm or nucleus morphology. H&E results are comparable with the results of the MTT assay. Figure 7 and Figure 8 show the changes generated by the microspheres and nanoparticles in 2D cell culture models. However, a possible improvement is the in vitro evaluation of cytotoxicity in 3D models. The 3D models mimic the tissue microenvironment. Authors suggest that microspheres proposed for antitumoral applications should also be tested in a tumor spheroid to elucidate the intracellular and intercellular signaling of cancer [45,46].

### 3.8. Hemolysis Assay

When the outer membrane of the erythrocytes is destroyed, hemoglobin is released. The lysis produces the release of the intracellular content of the red cells. The number of erythrocytes destroyed is estimated by measuring the amount of hemoglobin released in a sample. Hemolysis assays gives a reference to know the degree of toxicity of the nanoparticles that will be in contact with human blood. Concentrations from 1 to 10,000 µg/mL of bare magnetite nanoparticles and M-CS/Tween 80 microspheres were evaluated. The results were analyzed in accordance with the Standard Practice for Assessment of Hemolytic Properties of Materials ASTM F 756-08, which indicates that any material with a hemolysis rate of less than 2% is considered non-hemolytic [47]. On one hand, at a concentration of 10,000 µg/mL hemolysis produced by bare magnetite nanoparticles is higher than 2%, at this concentration it is considered a hemolytic material (Figure 9). Concentrations from 1 to 1000 µg/mL have a hemolysis rate lower than 2%, at those concentrations magnetite nanoparticles are considered a non-hemolytic material. On the other hand, M-CS/Tween 80 microspheres are considered non-hemolytic material at all concentrations tested. This experiment indicates that chitosan and Tween 80 improve the biological properties of magnetite nanoparticles.

### 3.9. Sister Chromatid Exchange

The formation of sister chromatid exchange (SCE) originated by DNA lesions has been a research subject for a long time. SCE is an exchange between segments of DNA in homologous loci from a chromosome. SCE formation is a normal cellular event, with a constant low frequency. An abnormal frequency of SCE is correlated with the production of chromosomal aberrations. The most accepted SCE formation mechanism is based on a double-strand break that occurs during DNA replication. Some environmental agents can cause a pathologic homologous recombination between the sister chromatids of chromosomes [48]. It is essential to evaluate if magnetite nanoparticles or M-CS/Tween 80 microspheres can cause this kind of exchange between genetic material because some changes in the chromosome sequence can cause several pathologies such as cancer. Turkez et al. evaluated the formation of SCE in cells exposed with nanoparticles of hydroxyapatite. They reported an average of 6.6 ± 1.4 SCE at concentrations of 100 µg/mL, and an average of 5.9 ± 1 SCE at concentrations of 10 µg/mL. They established that below 10 SCE, there is no significant damage in DNA or DNA-repair enzymes [49]. In this project, concentrations of 1, 10, and 100 µg/mL of magnetite nanoparticles and M-CS/Tween 80 microspheres were tested. Figure 10a shows an image of the chromosomes from one lymphocyte exposed to the positive control mitomycin C. Figure 10b shows the results of the number of SCEs generated in 50 samples for each different particle concentration tested. All the lymphocytes exposed with magnetite nanoparticles and with M-CS/Tween 80 microspheres shown less than 5 SCE per sample. In the positive control, the number of SCE was 23.38 ± 5.02, significantly different (*p =* 0.05) to all the tested samples. Results indicate that bare magnetite nanoparticles and M-CS/Tween 80 microspheres did not produce SCE chromosomal changes.

### 3.10. Damage in The Proliferation Mechanism

The proliferation index (PI) of the cultivated lymphocytes was analyzed using Equation (1)
PI = [M1 + 2(M2) + 3(M3)]/n(1)
where M1, M2, and M3 means the number of lymphocytes found in the first, second, and third cycle of mitosis, respectively; and n is the total of scored cells. Mitomycin C was used as a positive control of proliferation damage. Data was analyzed based of previous reports from different authors [50,51]. As Figure 11 shows, there was a significative difference (*p =* 0.05) between the positive control versus the negative control. None of the tested samples showed a significant difference (*p =* 0.05) against the negative control. Thus, results show that any particle tested caused changes in the normal cell cycle.

### 3.11. Protein Load

Microspheres were incubated with the BSA protein at different concentrations. The data analysis and interpretation were done based on other authors’ work [52,53]. Figure 12 shows that 500 µg of M-CS/Tween 80 microspheres could load onto their surface 900 µg of BSA protein. This result means that 1 µg of microspheres could load an average of 1.8 µg of BSA protein on their surface. These microspheres could be potentially redesigned to load different therapeutic proteins with similar physical properties than BSA.

## 4. Conclusions

In this study, we developed chitosan/Tween 80 microspheres loaded with magnetite nanoparticles, and these were characterized by FTIR, Raman, TGA, SEM, EDS, and DLS. The main point of this study was to determine the cell damage caused by the developed microspheres. The results confirm the safety of the use of chitosan/Tween 80 microspheres loaded with magnetite nanoparticles in vitro. There was no evidence of significant changes in mitochondrial metabolic activity, cell morphology, membrane lysis, sister chromatid exchanges, or the cell cycle. The authors recommend using a full set of assays to measure the cytotoxicity of nanoparticles because one single assay does not determine damage in different cell components. Although, these microspheres could be loaded with proteins with similar physical proprieties to BSA. For all the above, these materials represent a viable option in protein transport and release. The present study provides a fundamental understanding of the biological in vitro behavior of developed microspheres. This knowledge will open different research lines for the use of this microspheres as a target protein delivery system. The presented set of assays in this paper can help researchers to evaluate different nano and microparticles.

## Figures and Tables

**Figure 1 polymers-13-00400-f001:**
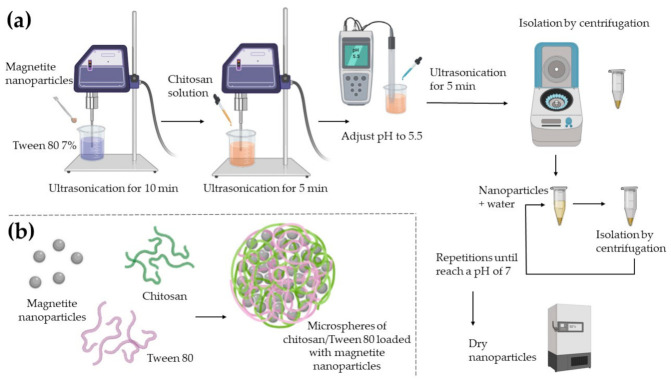
A scheme cartoon of the preparation of the microspheres. (**a**) A complete diagram of the methodology; (**b**) The interaction of chitosan and Tween 80 generate microspheres, which load the magnetite nanoparticles into a polymer network. Created with BioRender.com.

**Figure 2 polymers-13-00400-f002:**
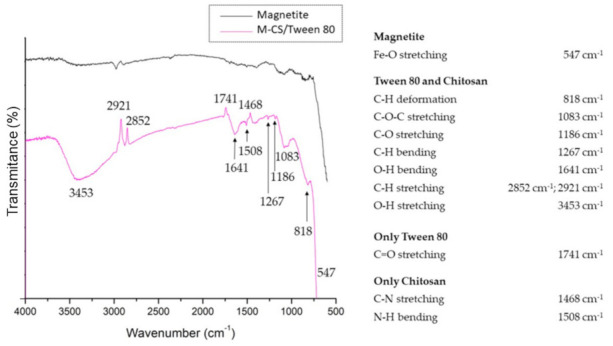
FTIR spectra of bare magnetite and chitosan/Tween 80 microspheres loaded with magnetite nanoparticles (M-CS/Tween 80).

**Figure 3 polymers-13-00400-f003:**
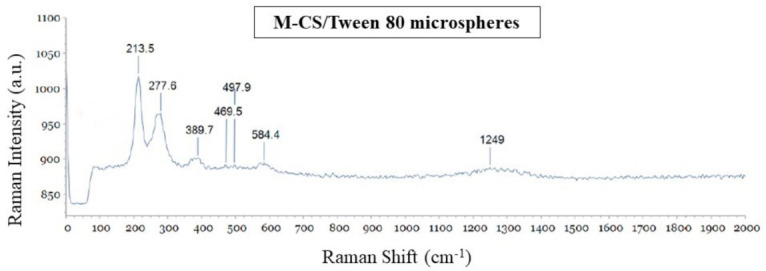
Raman spectrum of chitosan/Tween 80 microspheres loaded with magnetite nanoparticles (M-CS/Tween 80).

**Figure 4 polymers-13-00400-f004:**
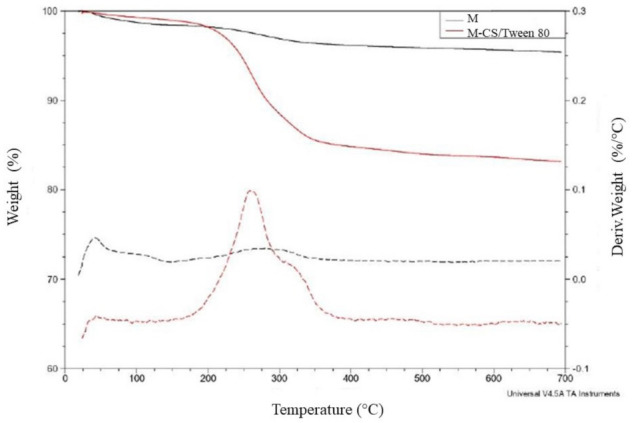
TGA spectrum. The weight loss of microspheres, from 250 to 350 °C, is given by the presence of chitosan and Tween 80.

**Figure 5 polymers-13-00400-f005:**
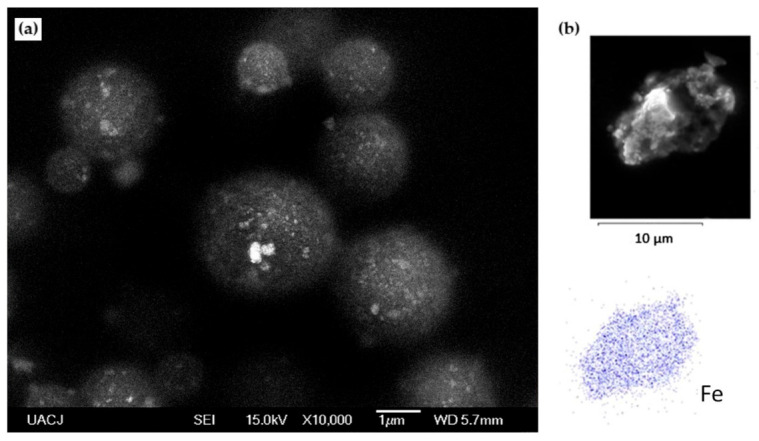
(**a**) SEM images of M-CS/Tween 80 microspheres at ×10,000; (**b**) EDS shows the distribution of Fe in the microsphere.

**Figure 6 polymers-13-00400-f006:**
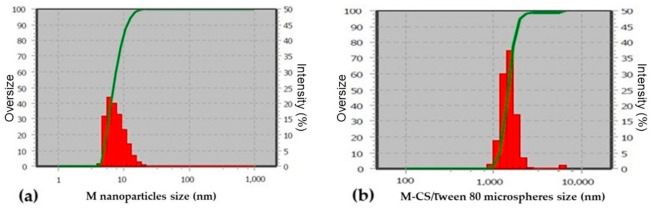
(**a**) Magnetite nanoparticle and (**b**) M-CS/Tween 80 microsphere size distribution. Polydispersity index (PDI) 0.77 and 0.22, respectively.

**Figure 7 polymers-13-00400-f007:**
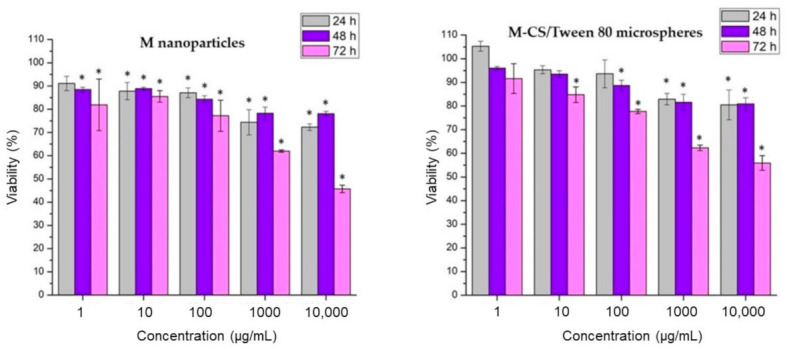
MTT assay of bare magnetite nanoparticles and chitosan/tween 80 microspheres loaded with magnetite nanoparticles. There is a significant difference * of particles cytotoxicity concerning negative controls.

**Figure 8 polymers-13-00400-f008:**
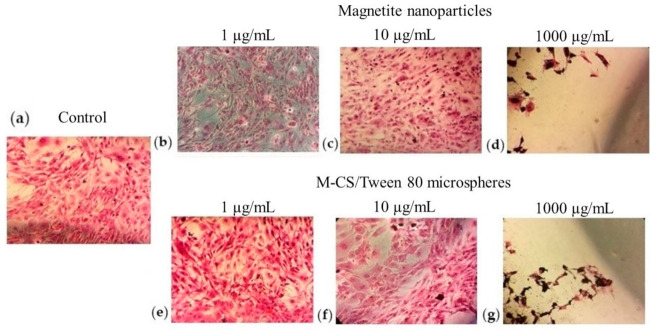
Optical microscope images of 40× of H&E staining at 24 h. (**a**) Negative control with no particles; (**b**–**d**) Magnetite nanoparticles and (**e**–**g**) M-CS/Tween 80 microspheres at different concentrations.

**Figure 9 polymers-13-00400-f009:**
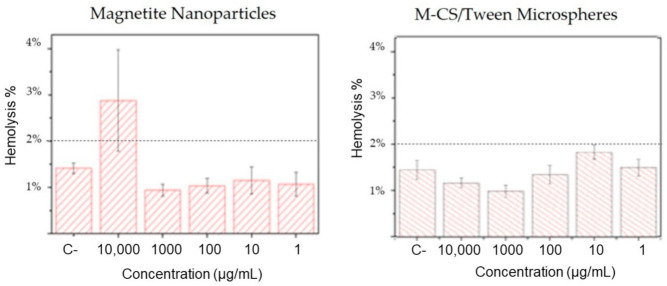
Hemolysis test results of bare magnetite nanoparticles and chitosan/tween 80 microScheme 2016. tested concentrations of 250, 500, and 3000 µg/mL of bare magnetite nanoparticles, reporting hemolysis rates below 2% at all concentrations [47]. In the present study, concentrations Figure 1 to 10,000 µg/mL of bare magnetite nanoparticles and M-CS/Tween 80 nanospheres were tested. Hemolysis rates obtained from magnetite nanoparticles are similar to the results of Macias et al.

**Figure 10 polymers-13-00400-f010:**
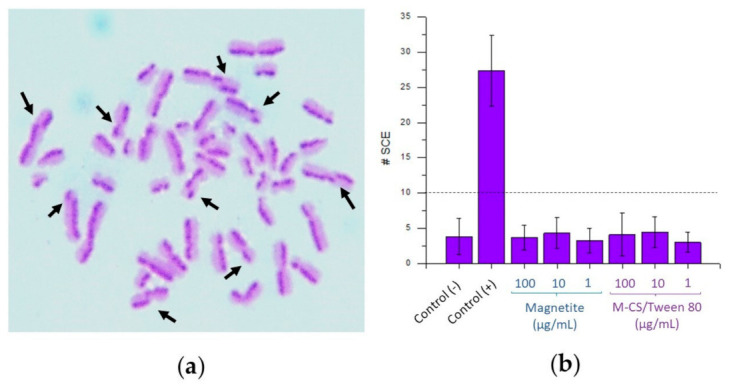
Sister chromatid exchange (SCE) analysis of lymphocytes exposed to microspheres. (**a**) An SCE positive sample (mitomycin C). The exchanges between one sister chromatid to another are witnessed because of the color. (**b**) The number of SCE found in all the samples.

**Figure 11 polymers-13-00400-f011:**
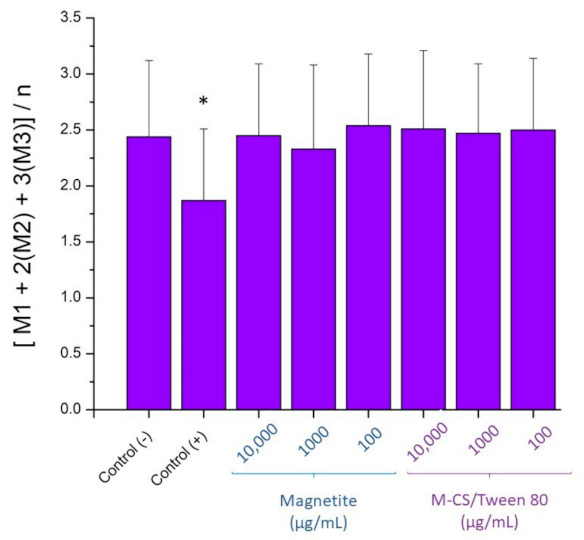
Proliferation index results of bare magnetite nanoparticles and chitosan/tween 80 microspheres loaded with magnetite nanoparticles. There was a significative difference (*p* = 0.05) between the positive control * versus the negative control.

**Figure 12 polymers-13-00400-f012:**
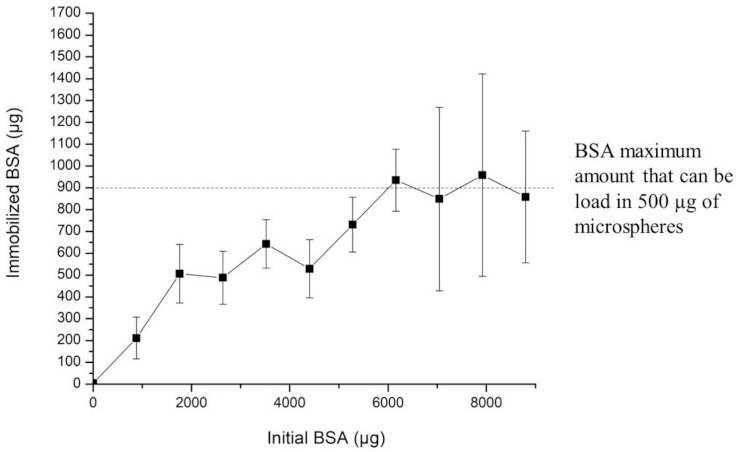
Different amounts of immobilized BSA are shown for different initial amounts of BSA.

## Data Availability

The data presented in this study are available on request from the corresponding author.

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
