# Peer review of "A Full Set of In Vitro Assays in Chitosan/Tween 80 Microspheres Loaded with Magnetite Nanoparticles"

_polymers, 2021, doi:10.3390/polym13030400_

Round 1
Reviewer 1 Report
The article entitled "A full set of in vitro assays on chitosan/tween 80 micospheres loaded with magnetite nanoparticles" (MNPs) reports on a thorough evaulation of the hemocompatibility of particulate drug carriers with focus on drug delivery in sensitive areas such as the human brain. The effects of MNPs and microcapsules loaded with MNPs at difference MNP concentration and exposure time on the viability and proliferation of different cell cultures are well characterized and documented.
The article is well written with a clear structure and a comprehensive analysis of the hemocompatibility of the suggested microcapsules-based approach for drug delivery.
I recommend publication of this article after considering the following minor revisions.
Line 90: Instead of "this methodology" it is better to say "the methodology described below" or "the methodology used in this report/article/research".
Add axis title to charts in figures 2, 6, and 7.
The main focus of this articles was on the qualification of safety of using the MNP-loaded microspheres. In a short section the authors added information about the capacity of these microspheres to adsorb (and probably carry) proteins by using BSA as a model protein. It would be interesting to extend this study to include the release of the proteins. Presumably such study is within scope of a future publication of the authors. In any case, some concluding remark on protein release should be added to the article.
Author Response
We appreciate you taking the time to review our manuscript. The observations allowed us to improve the final version. We will respond to each of your recommendations in a point-by-point manner.
Reviewer: The article entitled "A full set of in vitro assays on chitosan/tween 80 micospheres loaded with magnetite nanoparticles" (MNPs) reports on a thorough evaulation of the hemocompatibility of particulate drug carriers with focus on drug delivery in sensitive areas such as the human brain. The effects of MNPs and microcapsules loaded with MNPs at difference MNP concentration and exposure time on the viability and proliferation of different cell cultures are well characterized and documented.
The article is well written with a clear structure and a comprehensive analysis of the hemocompatibility of the suggested microcapsules-based approach for drug delivery.
I recommend publication of this article after considering the following minor revisions.
Answer: We thank you very much for the time you took to review the manuscript. We hope that the reported findings are of interest to the readers of Polymers journal.
Reviewer: Line 90: Instead of "this methodology" it is better to say "the methodology described below" or "the methodology used in this report/article/research".
Answer: Thanks for your recommendation. We have modified the text because we believe that it is better written in the way you suggested.
Reviewer: Add axis title to charts in figures 2, 6, and 7.
Answer: Thanks for your observation. We have added the axis titles to the indicated charts.
Reviewer: The main focus of this articles was on the qualification of safety of using the MNP-loaded microspheres. In a short section the authors added information about the capacity of these microspheres to adsorb (and probably carry) proteins by using BSA as a model protein. It would be interesting to extend this study to include the release of the proteins. Presumably such study is within scope of a future publication of the authors. In any case, some concluding remark on protein release should be added to the article.
Answer: Thanks for your recommendation. We have included the following concluding remark: For all the above, these materials represent a viable option in protein transport and release.
Reviewer 2 Report
The research article by Jorge et al. presents an interesting finding on the in vitro toxicity of magnetite nanoparticles loaded into chitosan/Tween 80 microspheres. The experimental section was well designed to evaluate the cell damage property of microspheres. The presentation of the findings was good. Hence, the article can be considered for publication with minor revision.
Suggested to include PDI values of magnetite nanoparticles and chitosan/Tween 80 microspheres along with the DLS.
Author Response
Dear reviewer, we thank you for reviewing the manuscript. We will respond to your comments in a point-by-point manner.
Reviewer: The research article by Jorge et al. presents an interesting finding on the in vitro toxicity of magnetite nanoparticles loaded into chitosan/Tween 80 microspheres. The experimental section was well designed to evaluate the cell damage property of microspheres. The presentation of the findings was good. Hence, the article can be considered for publication with minor revision.
Answer: We thank you very much for the time you took to review the manuscript. We hope that the reported findings are of interest to the readers of Polymers journal.
Reviewer: Suggested to include PDI values of magnetite nanoparticles and chitosan/Tween 80 microspheres along with the DLS.
Answer: Thanks for pointing out that we omitted the PDI values. The PDI values of magnetite nanoparticles and chitosan/Tween 80 microspheres were added in the text and in the Figure 6 caption.
Line 249: the polydispersity index (PDI) was 0.77 for magnetite nanoparticles and 0.22 for M-CS/Tween 80 microspheres.
Line 251: Figure 6. (a) Magnetite nanoparticle and (b) M-CS/Tween 80 microsphere size distribution. PDI 0.77 and 0.22, respectively.
Best regards,
Reviewer 3 Report
Authors proposed microspheres for different medical applications and presented their biological safety. The chitosan/Tween 80 micro-spheres loaded with magnetite nanoparticles were obtained and toxicology study was prepared for cell damage.
The publication is prepared exemplary way and deals with important chemical, biological and medical issues.
The results obtained in the work enable the further development of a new field of medicine consisting in the transport of drugs to specific places in living organisms with the use of an external alternating magnetic field.
The present study provides a fundamental understanding of the biological in vitro behavior of developed microspheres.
I recommend the paper in this form for publication due to the topicality of the obtained results and the essence of this research for the development of controlled methods of drug transport using magnetic carriers.
Author Response
Dear reviewer, we will respond to your comments in a point-by-point manner
Reviewer: Authors proposed microspheres for different medical applications and presented their biological safety. The chitosan/Tween 80 micro-spheres loaded with magnetite nanoparticles were obtained and toxicology study was prepared for cell damage.
The publication is prepared exemplary way and deals with important chemical, biological and medical issues.
The results obtained in the work enable the further development of a new field of medicine consisting in the transport of drugs to specific places in living organisms with the use of an external alternating magnetic field.
The present study provides a fundamental understanding of the biological in vitro behavior of developed microspheres.
Answer: We thank you very much for the time you took to review the manuscript. We hope that the reported findings are of interest to the readers of Polymers journal.
Reviewer: I recommend the paper in this form for publication due to the topicality of the obtained results and the essence of this research for the development of controlled methods of drug transport using magnetic carriers.
Answer: Thank you very much for recommending the paper for publication. We have made a great effort in this research and we believe that these findings will contribute to the development of materials for transporting and releasing proteins with pharmacological action.
This manuscript is a resubmission of an earlier submission. The following is a list of the peer review reports and author responses from that submission.
Round 1
Reviewer 1 Report
The article describes chitosan/Tween 80 nanoparticles with magnetite. The innovation of the article is weak because there have been many articles in this area. In addition, there are a few specific comments, as below.
- it is not clear what would be the utility of this type of microspheres
- What was the method for pulverization, the particles might break
- What is the particle size range after drying?
- Particles are huge size, much larger than 10 um. Also, what are the small particles on the large ones?
- Where are the positive and negative controls in Fig 5
- Not much difference in "a" and "e" of fig 6
- What was the incubation time for hemolysis? It is questionable how long the CS/Tween stays on the surface.
- Fig 9 is questionable because SD are too broad.
Reviewer 2 Report
SIGNIFICANCE OF THE WORK:
This manuscript reports a study regarding the cytotoxicity of microspheres loaded with magnetite nanoparticles.
TEXT:
The manuscript is, in general, well-crafted and written.
Please, correct the following typos:
L 48: “proprieties”
COMMENTS:
The issue discussed is well researched. However, the authors should comment on the benefits of the use of these microspheres instead of the nanoparticles beyond their intrinsic toxicity. How this new formulation can be attractive to be applied to real treatments. How stable are the microspheres? How is the interaction of the microspheres with macrophages?
Reviewer 3 Report
The Authors have submitted a manuscript regarding a cascade toxicity assays on magnetic micro-nanomaterials.
The claims are supported and the findings are interesting. The subject of the standard cascade assays is pivotal for the development of the next generation of nanotherapeutics and their translation to diseases management. Thus, I would suggest the acceptance of this manuscript after the following minors are addressed:
i) Introduction: Authors state "..few magnetic nanoparticles have reached preclinical trials..". That's not true. Indeed, some magnetic nanomaterials reached the marked (for example, Endorem). The Authors should re-phrase and look at specific literature (as for example, doi: 10.1002/wnan.1416)
ii) The Authors reported assays performed on 2D cultures. They should discuss in the ms the employment of 3D models for the production of more reliable and translatable findings (for example, doi: 10.1016/j.apmt.2019.100552)
Round 2
Reviewer 2 Report
According to authors’ answers to my comments, I gather that the study presented is rather preliminary and more work must be conducted. Furthermore, in my opinion, a fully functional system should be studied instead of simple nanoparticles without any drug loading.